# Bandgap Characteristics of Boron-Containing Nitrides—Ab Initio Study for Optoelectronic Applications

**DOI:** 10.3390/ma17205120

**Published:** 2024-10-21

**Authors:** Pawel Strak, Iza Gorczyca, Henryk Teisseyre

**Affiliations:** 1Institute of High Pressure Physics, Polish Academy of Sciences, 01-142 Warsaw, Poland; iza@unipress.waw.pl (I.G.); teiss@ifpan.edu.pl (H.T.); 2Institute of Physics, Polish Academy of Sciences, 02-668 Warsaw, Poland; 3Université Côte d’Azur, CNRS, Centre de Recherche sur l’Hétéro-Epitaxie et ses Applications, 06-905 Valbonne, France

**Keywords:** random alloy, BInN, BGaN, BAlN, bulk materials

## Abstract

Hexagonal boron nitride (h-BN) is recognized as a 2D wide bandgap material with unique properties, such as effective photoluminescence and diverse lattice parameters. Nitride alloys containing h-BN have the potential to revolutionize the electronics and optoelectronics industries. The energy band structures of three boron-containing nitride alloys—B*_x_*Al_1−*x*_N, B*_x_*Ga_1−*x*_N, and B*_x_*In_1−*x*_N—were calculated using standard density functional theory (DFT) with the hybrid Heyd–Scuseria–Ernzerhof (HSE) function to correct lattice parameters and energy gaps. The results for both wurtzite and hexagonal structures reveal several notable characteristics, including a wide range of bandgap values, the presence of both direct and indirect bandgaps, and phase mixing between wurtzite and hexagonal structures. The hexagonal phase in these alloys is observed at very low and very high boron concentrations (*x*), as well as in specific atomic configurations across the entire composition range. However, cohesive energy calculations show that the hexagonal phase is more stable than the wurtzite phase only when *x* > 0.5, regardless of atomic arrangement. These findings provide practical guidance for optimizing the epitaxial growth of boron-containing nitride thin films, which could drive future advancements in electronics and optoelectronics applications.

## 1. Introduction

Since the breakthrough studies of quantum transport in graphene [1], there has been extraordinary interest in layered materials, particularly nanostructures composed of single layers of van der Waals (VdW) crystals, such as transition metal dichalcogenides (TMDs), graphene, and boron nitride. This list has since expanded to include franckeite, a naturally occurring van der Waals heterostructure; black phosphorus, a *p*-type 2D material; niobium diselenide, a 2D superconductor with a critical temperature of T_c_ = 7.2 K; and chromium (III) bromide, a ferromagnetic 2D semiconductor. These systems are promising platforms for developing various two-dimensional (2D) devices [2], with potential applications in electronics, optoelectronics, single-photon detectors, catalysis, sensors, and flexible electronics.

Boron and nitrogen, located in groups III and V of the periodic table, respectively, combine to form boron nitride (BN), which exists in several polymorphs: hexagonal (h-BN), cubic (c-BN), wurtzite (w-BN), amorphous (a-BN), and rhombohedral (r-BN). Each polymorph has unique properties. However, only hexagonal boron nitride (h-BN) is considered a true 2D material, possessing both a direct wide bandgap (~6 eV) and an indirect bandgap (~4.5 eV). This wide bandgap contributes to h-BN’s excellent electrical insulating properties, making it useful in high-temperature and high-frequency electronic applications.

Moreover, h-BN exhibits strong exciton recombination at energies of 5.76 eV and 5.86 eV, attributed to direct exciton recombination supported by phonon emission. These properties position h-BN as a promising material for future optoelectronic applications [3].

The photoluminescence (PL) intensity of h-BN layers has been found to be nearly 100 times greater than that of commercial AlN layers. This remarkable PL efficiency is attributed to two key factors: high exciton binding energy and strong electron-phonon coupling (Fröhlich interaction). These unique optical properties have even enabled the demonstration of optically pumped laser action in high-purity h-BN crystals [4].

In addition, h-BN is an excellent insulating material for use as a barrier in nanostructures composed of van der Waals crystals and transition metal dichalcogenides (TMDs). Numerous PL studies on TMD monolayers have shown that free exciton emission linewidths are subject to considerable inhomogeneous broadening, typically caused by variations in the substrate’s local spatial properties, strain, and the presence of adsorbed atoms or molecules on the surface [5,6,7,8]. A significant improvement in the emission spectra of TMD layers can be achieved by hermetically sealing them between exfoliated h-BN films. This encapsulation enhances the emission spectra of h-BN–encapsulated TMD monolayers compared to non-encapsulated counterparts, allowing for more detailed studies of various excitonic features.

BN can also be more effectively *p*-type doped than AlN [9]. The combination of *p*-type BN and AlGaN alloys with high aluminum content is particularly promising [10], offering new possibilities in optoelectronics, especially for deep UV and high-power electronic applications.

However, the focus of our calculations is not on h-BN itself but on mixed tri-component alloys with indium, gallium, and aluminum, which offer tunability of the bandgap and related light emission properties. Recent advancements in using h-BN-based structures for deep UV photonic devices have been underscored by the successful synthesis of hexagonal BGaN/BN layered quantum wells, which demonstrate promising PL emissions and favorable electrical properties.

Despite the growing interest in h-BN and the reported experimental progress [9,10], there remains limited theoretical exploration of this material. Even the nature of the bandgap in h-BN alloys is still a subject of debate [3,4]. However, supporting ongoing experimental work in laboratories worldwide with a solid theoretical foundation is crucial for advancing the application of quantum structures based on h-BN in deep UV optoelectronics.

To address this need, we will focus on the electronic band structure of hexagonal alloys—h-B*_x_*Al_1−*x*_N, h-B*_x_*Ga_1−*x*_N, and h-B*_x_*In_1−*x*_N—across the full range of compositions (*x*). With their high bandgap values and a broad range of lattice parameters, these nitride alloys have the potential to revolutionize the electronics and optoelectronics industry in the near future.

A schematic representation of the bandgap dependence on alloy composition and lattice constant for these alloys is presented in Figure 1.

Despite strong motivation, the growth of high-quality nitride alloys containing BN, in both wurtzite and hexagonal phases, presents significant challenges due to the considerable lattice mismatch between BN and other III-N compounds. The larger lattice mismatch between the binary compounds—approximately 20% for BN/GaN compared to 18% for BN/AlN—makes combining BN with GaN even more challenging than with AlN.

The significant lattice mismatch between BN and AlN, as well as GaN, leads to large miscibility gaps. As a result, these alloys have predominantly been studied in the dilute regime of boron incorporation, owing to the inherent difficulties in achieving a uniform mixture across the full range of boron content.

The MOVPE (metal–organic vapor phase epitaxy) technique remains the most successful growth method for classical nitrides, so it is not surprising that it has been applied to the growth of BAlN and BGaN alloy systems. Despite the large lattice mismatch between BN and AlN, secondary-ion mass spectrometry (SIMS) measurements have confirmed the incorporation of up to 17% aluminum in layered BAlN alloy systems [21]. The same group was even able to produce Distributed Bragg Reflectors [22]. However, other reports using the same technique have shown significantly lower aluminum concentrations [23]. For the BGaN system, the highest reported gallium concentration, up to 7%, was documented by Wang in his PhD thesis [24].

High boron doping in GaN and AlN layers is a more common approach, and considerable efforts have been made in this area. The highest reported boron concentration for BAlN films was by Polyakov et al. [25], who found that single-phase wurtzite (WZ) BAlN films could only be grown with boron compositions not exceeding 10%. The same author also reported relatively high boron concentrations, up to 7%, in BGaN layers [26]. Progress related to increasing boron concentrations in GaN has been thoroughly described by Możdżyńska et al. [27], while both BAlN and BGaN alloy systems, along with their stability issues, are covered in a comprehensive review by Kudrawiec et al. [28].

The lattice mismatch between BN and InN is even greater, at approximately 28%. As a result, the epitaxial growth of wurtzite or hexagonal BInN alloys will be exceptionally difficult, if not impossible. To our knowledge, there are currently no reports of either experimental or theoretical studies of BInN in the literature.

The existing theoretical and experimental data indicate a highly complex nature of h-BAlN and h-BGaN alloys concerning their bandgaps and crystallographic structures. Moreover, the reported data vary significantly, suggesting different composition ranges where the hexagonal phase is stable [28,29,30,31,32]. Therefore, we have decided to conduct a comprehensive study of the bandgap behavior of all three alloys—B*_x_*Al_1−*x*_N, B*_x_*Ga_1−*x*_N, and B*_x_*In_1−*x*_N—in both hexagonal and wurtzite phases, including various atomic arrangements within the unit cell.

In the next section, we present the methods of calculation. Section 3 provides a description of the results and discussion, focusing on direct and indirect bandgaps, the wide range of their values, and significant bowings. We also examine the effects of different atomic arrangements (clustering) and the impact of hexagonal-wurtzite phase mixing. Section 4 summarizes our findings.

## 2. Methods of Calculations

Most of the ab initio calculations were performed using the commercial Vienna Ab Initio Simulation Package (VASP) version 6.4.2, developed by the University of Vienna, Austria [33,34,35,36]. This density functional theory (DFT) code uses a plane-wave basis set to solve the Kohn–Sham equations. The plane-wave functions are characterized by the momentum vector values $\vec{k}$. The maximum value of the momentum vector is determined by the energy cutoff, which is defined by the maximum kinetic energy cutoff, $E_{cut}=\frac{\hbar^{2}k^{2}}{2m}$. The density of the $\vec{k}$ points is determined by the system size $\left(L_{i},\;i=x,\;y,\;z\right)$ through the period boundary conditions (PBC), given by $\left.k_{i}=\frac{2\pi }{L_{i}}\right.$. The same PBC is used to solve the coupled Poisson equations using a Fourier series. In this work, the energy cutoff was set to $E_{cut}=400\;eV$.

The plane-wave basis set for the all-electron solution of systems containing metal atoms, such as boron, aluminum, gallium, indium, and nitrogen, is prohibitively large. As a result, even for relatively small system sizes, a reduction in the basis is necessary. To address this, the electron sets of all atoms are divided into two distinct classes.

The first set consists of the atomic core electrons, which are not explicitly considered in the calculations. This set includes closed-shell electrons that are only marginally affected by crystal bonding. Therefore, the atomic cores are treated as “frozen”, and only polarization effects are accounted for through minor corrections.

The second set, which is explicitly considered, consists of the valence electrons. These electrons represent the total number of electrons in the simulation cell. This separation necessitates a special formulation where the Coulomb potential is replaced by a more computationally efficient approach, such as the use of a regular function or a set of matrix elements.

In VASP, norm-conserving or projector-augmented wave (PAW) potentials, generated by Kresse, are available for these calculations [37,38]. The standard ab initio method results for semiconductors tend to be inaccurate, as they typically underestimate energy bandgaps by about 40% compared to experimental observations [39]. To address this issue, the standard DFT functional is supplemented with the Heyd–Scuseria–Ernzerhof (HSE) functional, which augments the standard DFT approach with a Hartree–Fock exchange correction [40]. While this implementation is computationally expensive, it is well-suited for small simulated systems. As a result, the energy gaps of nitride-based semiconductors can be predicted with relatively high accuracy using the hybrid HSE functional. Experimental data are frequently employed to verify the quality of the parameterization. In this study, we compare the lattice parameters obtained from ab initio DFT calculations for bulk nitrides, as well as their bandgaps calculated with the HSE correction, against available experimental data. All relevant data are summarized in Table 1.

Our calculated lattice parameters were compared with those measured by X-ray diffraction, and the agreement between the two is generally very good. Even for BN, the results are quite satisfactory, considering the significant challenges associated with its synthesizing.

Experimental data on the bandgap value for h-BN are scarce, but our calculated value aligns well with the experimental result reported in [3]. Similarly, the agreement between our calculated and experimental bandgap values for AlN (aluminum nitride) and GaN (gallium nitride) is also quite good. The optical bandgap of InN has been the subject of considerable debate, with the consensus value now established at 0.65 eV [18,19,20]. Our calculated value is slightly larger, at 0.9 eV.

## 3. Results and Discussion

We performed several sets of energy band structure calculations for three boron-containing nitride alloys, h-B*_x_*Al_1−*x*_N, h-B*_x_*Ga_1−*x*_N, and h-B*_x_*In_1−*x*_N, in the hexagonal structure and in the wurtzite structure (w-B*_x_*Al_1−*x*_N, w-B*_x_*Ga_1−*x*_N, and w-B*_x_*In_1−*x*_N). These calculations were conducted for various boron contents and different atomic geometries, ranging from random to clustered configurations. The resulting dependencies of the bandgaps on boron content are shown in Figure 2a–c for all three alloys.

To simulate a random distribution of atoms, we employed 128-atom supercells, while more clustered atomic configurations were modeled using 16-atom or 32-atom supercells. Direct and indirect bandgaps are represented by open and filled symbols, respectively. The results reveal several interesting effects, which will be discussed in the following sections.

The resulting bandgaps are either direct or indirect, represented by open and filled symbols, respectively, in Figure 2. The bandgap values exhibit a wide range for a given boron content, depending on the geometric arrangement of the atoms in the supercell. These and other effects related to bandgap behavior will be discussed in detail in the following sections.

### 3.1. Direct and Indirect Bandgaps

Unfortunately, there are not many publications on the hexagonal phase of boron-containing nitrides. Therefore, the comparison with the reports of other authors is limited to the wurtzite structure. What has been published on this subject is mainly theoretical work and the results are sometimes quite different. For wurtzite B*_x_*Al_1__−_*_x_*N, other ab initio DFT calculations indicate that direct to indirect bandgap transitions occur at *x*~0.12) [41] or at *x*~0.28 [42]. The latter value is in perfect agreement with our first region (*x* = 0.29), with the note that in the next region, we can still observe direct bandgaps for certain atomic configurations. For the w-B*_x_*Ga_1__−_*_x_*N structure, another first-principles calculation indicates that a transition from direct to indirect bandgap occurs for B concentrations *x* > 0.5 [43]. This value is more comparable to our value of *x* = 0.65, where all bandgaps in w-B*_x_*Ga_1__−_*_x_*N are already indirect.

In both wurtzite and hexagonal crystallographic structures, the fundamental bandgap of BN is indirect, while AlN exhibits a direct bandgap. For boron alloys in both structures, the nature of the bandgap—whether direct or indirect—depends on the alloy composition and atomic arrangement. For each alloy, we can categorize the entire range of *x* into three regions: from 0 to *x*_1_, from *x*_1_ to *x*_2_, and from *x*_2_ to 1. In w-B*_x_*Al_1__−_*_x_*N *x*_1_~0.29, *x*_2_~0.72, in w-B*_x_*Ga_1__−_*_x_*N *x*_1_~0.12, *x*_2_~0.65, and in w-B*_x_*In_1__−_*_x_*N *x*_1_~0.10, *x*_2_~0.68.

In the first range, all bandgaps are direct; in the second range, both direct and indirect bandgaps appear; and in the final range, all bandgaps are indirect. These three regions are illustrated in Figure 2 for each alloy, separated by dashed vertical lines. Comparing the wurtzite and hexagonal structures, it is evident that the second range (where both direct and indirect gaps occur) is generally shorter in the hexagonal structure than in the wurtzite structure.

Unfortunately, the literature is limited on the hexagonal phase of boron-containing nitrides, restricting comparisons to the wurtzite structure. Most available publications on this topic are theoretical, and the results can vary significantly. For wurtzite B*_x_*Al_1__−_*_x_*N, other ab initio DFT calculations suggest that direct-to-indirect bandgap transitions occur at *x* ≈ 0.12 [41] or *x* ≈ 0.28 [42]. The latter value aligns closely with our first region (*x* = 0.29), noting that direct bandgaps may still be observed in the next region for certain atomic configurations.

For the wurtzite B*_x_*Ga_1__−_*_x_*N structure, another first-principles calculation indicates that the transition from direct to indirect bandgap occurs for boron concentrations *x* > 0.5 [43]. This value is comparable to our result of *x* = 0.65, where all bandgaps in w-B*_x_*Ga_1__−_*_x_*N are already indirect.

With respect to B*_x_*In_1−*x*_N, there are no reports in the literature of experimental or theoretical studies of this alloy, either in the wurtzite or hexagonal phase. We found only one paper dealing with the cubic phase of this alloy [44]. The authors of this paper found a direct–indirect bandgap transition at *x* = 0.83. This value is somewhat similar to our result for the wurtzite phase (*x* = 0.70).

### 3.2. Large Spread of Bandgap Values

A significant spread of bandgap values, ΔE_g_, for a given boron composition *x* was observed in all three alloys considered. The largest ΔE_g_, approximately 2.3 eV, was found for *x* = 0.25 in h-B*_x_*Al_1__−_*_x_*N. In h-B*_x_*Ga_1__−_*_x_*N and h-B*_x_*In_1__−_*_x_*N, the spread is smaller, around 1.5 eV at *x* = 0.5.

In the wurtzite phase, ΔE_g_ is generally smaller, ranging from 1.2 to 1.5 eV, and occurs for 0.5 < *x* < 0.6. A similar, though slightly smaller, bandgap spread has also been observed in previously studied alloys. The maximum theoretical values of ΔE_g_ were approximately 0.5 eV in InGaN [45], 1.2 eV in InAlN [45], and 1.3 eV in ZnMgO [46].

The primary reason for the wide range of bandgap values in these alloys is their dependence on atomic arrangement—specifically, whether the atoms are uniformly distributed throughout the alloy or more clustered. This same effect appears to contribute to the large spread of bandgaps observed in boron-containing alloys.

### 3.3. Clustering

In indium-containing nitride alloys [45] and zinc-containing oxides [46], the bandgaps were significantly lower when the atoms were clustered (i.e., grouped in a part of the supercell or in a specific layer of the structure) compared to when they were uniformly distributed.

In our calculations, we used large 128-atom supercells to achieve a random configuration of atoms. For configurations where B atoms are more or less clustered, it was more convenient to use smaller supercells (16 and 32 atoms). The first observation is that the bandgap values corresponding to the clustered atomic configurations are located in the lower region of the energy gaps, consistent with previous findings [45,46]. In Figure 2, the two curves representing the bandgap values for a given alloy and structure show that the upper curve corresponds to a random alloy, while the lower curve corresponds to a clustered atomic configuration.

For a more detailed analysis, we consider the case of h-B*_x_*Al_1__−_*_x_*N (Figure 2a). The two lower energy bandgaps, the lowest at E_g_ = 2.75 eV and a slightly higher value of E_g_ = 2.98 eV for *x* = 0.375, correspond to the clustered configurations for this composition. The atomic structures of these alloys are illustrated in Figure 3.

In the first configuration (Figure 3a), all three boron atoms are positioned in a superlattice layer, while the alloy with the slightly higher E_g_ (Figure 3b) has two boron atoms in one layer and one boron atom in the adjacent layer. As a result of this reduced clustering, the resulting bandgap increases slightly to E_g_ = 2.98 eV.

In the case of InGaN and AlGaN alloys, the significant decrease in E_g_ in clustered configurations is attributed to the shortening of the In-N bonds in the In-containing alloys compared to those in pure InN. For example, in the clustered In_0_._25_Al_0_._75_N alloy, the In-N bond length is approximately 2.02 Å, while in pure InN, it measures about 2.15 Å (see Figures 10 and 11 in [45]). This bond shortening results in stronger interactions at the top of the valence band between states originating from In and the nearest N atoms, effectively pushing the top of the valence band upward and significantly reducing the bandgap.

To investigate whether an analogous effect occurs in the h-B*_x_*Al_1−*x*_N alloy, we calculated the partial density of states (PDOS) for both uniform and clustered atomic configurations. We selected the lowest and highest points in Figure 2a for *x* = 0.375, with bandgaps of E_g_ = 2.75 eV and E_g_ = 4.46 eV, respectively. The lowest point corresponds to the most clustered case, while the highest point represents the structure with randomly distributed atoms.

The obtained PDOS for these two cases is shown in Figure 4a,b. The difference in bandgap values for these configurations is attributed to the variation in the shape of the nitrogen contribution to the PDOS at the top of the valence band (VB). Specifically, the distinct shape of the nitrogen contribution observed in Figure 4a leads to a decrease in the bandgap due to the stronger interaction between the boron and nitrogen states in the clustered configuration.

It is noteworthy that the B-N bond lengths (approximately 1.6 Å) in the clustered atomic configuration are generally shorter than those in the uniform case (up to approximately 1.7 Å). This effect is analogous to the observations made in InGaN and AlGaN alloys [45].

### 3.4. Bandgap Bowings

Another characteristic parameter describing bandgap behavior is the bandgap bowing b. Our calculated bandgap bowings are significantly higher for clustered cases than for uniform distributions of atoms. Additionally, they are greater for the hexagonal phase than for the wurtzite phase.

In w-B*_x_*Al_1__−_*_x_*N, the bowing for the uniform wurtzite case is approximately b ≈ 1.7 eV, while for the clustered case, it reaches b ≈ 7.2 eV, with an average bowing of about 4.4 eV. In h-B*_x_*Al_1__−_*_x_*N, the bowing is significantly higher, ranging from 3.78 eV to 12.4 eV (average: b ≈ 8.5 eV). In w-B*_x_*Ga_1__−_*_x_*N, the average bowing is about 3.1 eV, while for the hexagonal phase, it is around 8.2 eV. Notably, the average values of bandgap bowing in B*_x_*Al_1__−_*_x_*N and B*_x_*Ga_1__−_*_x_*N in the wurtzite phase (3*–*4 eV) and in the hexagonal phase (approximately 8 eV) are similar.

Hybrid DFT calculations [42] indicate that the direct gap bowing parameter of the w-B*_x_*Al_1__−_*_x_*N alloy varies from b = 8.38 eV at very small *x* to b = 8.67 eV at *x* = 0.17. Bandgap bowing in thin films of w-B*_x_*Ga_1__−_*_x_*N has been studied experimentally in [47]. Using optical transmission, photoluminescence, and X-ray diffraction, a significant bandgap bowing of b = 9.2 eV was observed. Through first-principles calculations with a hybrid function, Turiansky et al. [43] found a bowing parameter of about 8.68 eV for w-B*_x_*Ga_1−*x*_N alloys with boron concentrations below 20%.

In our calculations, the effect of bandgap bowing is most pronounced in B*_x_*In_1−*x*_N (see Figure 2c), with a higher value obtained for the clustered case in the hexagonal phase. This result aligns with the chemical trends in the bowing parameter describing III-N ternary alloys, which show an increase in the bowing parameter with increasing lattice mismatch. Following this trend, the bowing parameter for the direct and indirect bandgaps in B*_x_*In_1−*x*_N was estimated to be b ≈ 11 eV [28].

### 3.5. Hexagonal-Wurtzite Phase Transitions

Figure 2 illustrates the energy regions covered by the bandgaps calculated for the wurtzite and hexagonal structures. However, the situation is more complex than depicted. When calculating the electron band structure of the alloys in the hexagonal phase, we observe that the bandgaps located in the higher energy region (near the “wurtzite region” or in the overlap region between the hexagonal and wurtzite phases) tend to exhibit a wurtzite or wurtzite-like structure. This effect arises from the strong lattice relaxation characteristic of boron-containing nitrides.

Notably, the substantial lattice relaxation that leads to the transition from the hexagonal to the wurtzite phase occurs in the overlapping regions of the hexagonal and wurtzite phases or very close to them. The pure hexagonal phase is observed only for very low x values (i.e., *x* < 0.1) and for high values of *x* (i.e., *x* > 0.7 in h-B*_x_*Al_1−*x*_N and h-B*_x_*Ga_1−*x*_N and *x* > 0.9 in h-B*_x_*In_1−*x*_N). Additionally, the lowest bandgaps corresponding to clustered arrangements of atoms, which remain outside the overlapping regions for the entire range of compositions, are characterized by a hexagonal structure.

The effect of the hexagonal-to-wurtzite phase transition at the boundary between the two phases is illustrated in Figure 5a, using the example of the h-B*_x_*Al_1−*x*_N alloy with *x* = 0.375 and E_g_ = 4.46 eV (shown in Figure 3a as the highest point in a series of bandgaps for *x* = 0.375, close to the “wurtzite region”). For comparison, the structure of pure wurtzite BN is shown in Figure 5b, and pure hexagonal BN is shown in Figure 5c. It is clear from Figure 5a that the structure displayed has a “mixed” character, with more wurtzite-like than hexagonal features.

Similar conclusions regarding hexagonal-to-wurtzite (or vice versa) phase transitions have been reported in the literature. Zhang et al. [29], based on formation energy calculations, divided the entire composition region into three intervals: (1) *x* = 0 to 0.5, (2) *x* = 0.5 to 0.81, and (3) *x* = 0.81 to 1, each with its own distinct trend (see Figure 4 in [29]). The first and second intervals correspond to the wurtzite-B*_x_*Al_1−*x*_N (w-B*_x_*Al_1−*x*_N), while the third region corresponds to the hexagonal-B*_x_*Al_1−*x*_N (h-B*_x_*Al_1−*x*_N). This division is in agreement with our observations. The phase transition from wurtzite to hexagonal observed at *x* = 0.81 in [16] can be compared to our estimated transition value of *x* > 0.7.

A transmission electron microscopy (TEM) study [30] further revealed the presence of wurtzite, Al-rich B*_x_*Al_1−*x*_N phases embedded within a matrix of layered hexagonal h- B*_x_*Al_1−*x*_N in the composition range *x* > 0.83.

The complex nature of the bandgap behavior in B*_x_*Al_1−*x*_N has been confirmed in the work of Milne et al. [31]. Their ab initio calculations for w-B*_x_*Al_1−*x*_N show that its structure deviates from wurtzite symmetry in certain *x* ranges due to the large lattice mismatch between AlN and BN. The B*_x_*Al_1−*x*_N structures predominantly exhibit a tetrahedral sp3 bonding environment; however, some structures show sp2 bonding, similar to hexagonal BN. For example, the structure at *x* = 0.417 was found to be characterized by sp2 bonding, corresponding to h-BN.

Another theoretical study [32] on both B*_x_*Al_1−*x*_N and B*_x_*Ga_1−*x*_N alloys also confirms that the layered hexagonal phase, characteristic of h-BN, becomes more dominant at very high boron compositions.

These theoretical results have been experimentally confirmed for h-B*_x_*Ga_1−*x*_N in [48], where layered h-B*_x_*Ga_1−*x*_N was observed at boron compositions exceeding 92%. Using an h-BN epilayer as a template, they successfully synthesized h-B*_x_*Ga_1−*x*_N alloys and quantum wells (QWs) by metal–organic chemical vapor deposition (MOCVD), achieving a maximum gallium composition of about 7% (*x* > 93).

There are currently no theoretical or experimental data on h-B*_x_*In_1−*x*_N in either the wurtzite or hexagonal structures. Our findings indicate that, in h-B*_x_*In_1−*x*_N, nearly all the bandgaps fall within a region common to both hexagonal and wurtzite phases. Only the lowest hexagonal bandgaps and the highest wurtzite bandgaps fall outside this region. The bandgaps within the common region are primarily associated with the “mixed” hexagonal-wurtzite structure. An example of such a mixed structure, corresponding to *x* = 0.75 and E_g_ = 2.53 eV in Figure 2c, is shown in Figure 6.

Based on our considerations regarding the bandgap characteristics, atomic geometries, and crystallographic structures, we can conclude that the pure hexagonal phase is observed only at very low and very high values of *x*, as well as for the lowest bandgaps, which correspond to a clustered arrangement of atoms. To verify this conclusion, we decided to estimate the stability of the investigated structures.

### 3.6. Stability of the Structures

One of the most fundamental properties of crystalline systems is their cohesive energy, which plays a crucial role not only in bulk materials like random alloys but also in the formation of structures such as 2D-layered materials and nanorods [49,50].

To estimate the composition range where the hexagonal phase is more stable than the wurtzite phase, we calculated the cohesive energies for all three alloys. The results for both hexagonal and wurtzite phases, across all the atomic arrangements considered, are shown in Figure 7a–c for B*_x_*Al_1−*x*_N, B*_x_*Ga_1−*x*_N, and B*_x_*In_1__−_*_x_*N, respectively. The red and blue solid lines represent fits to the lowest cohesive energy points for the hexagonal and wurtzite phases, respectively. We found that these points correspond predominantly to random atomic arrangements, whereas the highest red points correspond to the most clustered configurations. This indicates that a uniform (or random) atomic arrangement is more stable than a clustered one.

The calculations show that the hexagonal phase can be more stable than the wurtzite phase for boron contents of *x* > 0.65 in B*_x_*Al_1−*x*_N and B*_x_*Ga_1−*x*_N and for *x* > 0.3 in B*_x_*In_1−*x*_N. These findings suggest that the range of compositions where the hexagonal phase is stable is limited to high boron contents in all three alloys considered.

## 4. Discussion and Summary

First-principles energy band structure calculations have been conducted for three boron-containing nitride alloys: B*_x_*Al_1__−*x*_N, B*_x_*Ga_1__−*x*_N, and B*_x_*In_1__−*x*_N in both wurtzite and hexagonal crystallographic structures. In general, our results are in qualitative agreement with previous theoretical and experimental studies. However, some discrepancies in the bandgap properties and alloy composition dependence have been reported by different researches. Our findings indicate that boron-containing nitride alloys exhibit highly interesting properties, as follows:A significant spread in the bandgap values, especially in the hexagonal phase (ranging from 1.5 eV to 2.3 eV), was observed, which is related to the different arrangements of atomic positions in the supercell. Unlike other studies, our work includes an analysis of the dependence of bandgap values on the specific arrangement of atoms in the crystal lattice. When comparing this large variation in E_g_ values to the scatter in reported experimental data, it appears that, in addition to other nanoscale factors, different degrees of clustering in samples grown in various laboratories may also contribute to these discrepancies;Both direct and indirect bandgaps were observed. In both the wurtzite and hexagonal phases, the fundamental bandgap of BN is indirect, while AlN, GaN, and InN are direct bandgap semiconductors. For very small amounts of boron (low values of *x*), all the obtained bandgaps are direct, adopting the direct bandgaps of the nitrides. However, for *x* > 0.1, indirect gaps also begin to appear, depending on the atomic configurations. For very large boron concentrations (*x* > 0.7), the bandgap adopts the character of the BN bandgap, and all gaps become indirect;Mixing of wurtzite and hexagonal phases. In regions where the wurtzite and hexagonal phases overlap or are close to each other (see Figure 3), we observe transitions from the intended hexagonal phase to a wurtzite-like or “mixed” phase due to lattice relaxation. The pure hexagonal phase in h-B*_x_*Al_1−*x*_N and h-B*_x_*Ga_1−*x*_N is observed only at very low or very high *x* values or in cases where the atoms exhibit more clustered arrangements across the composition range. However, stability calculations for all three alloys under consideration restrict the hexagonal phase to high boron compositions. The existence of the hexagonal phase at high boron compositions in BAlN and BGaN has also been reported in the literature. For instance, the hexagonal phase of B*_x_*Al_1−*x*_N for *x* > 0.81 was theoretically predicted in [29] and experimentally confirmed in [30] for *x* > 0.83. Another theoretical study [32] on both B*_x_*Al_1−*x*_N and B*_x_*Ga_1−*x*_N alloys also confirmed that the layered hexagonal phase dominates at very high boron compositions. Milne et al. [31] pointed out the existence of different phases depending on the B*_x_*Al_1−*x*_N composition, observing both sp3 and sp2 bonding environments in certain structures. These effects were further described in reviews [28,51]. Additionally, B*_x_*Ga_1−*x*_N has been reported as metastable in some regions, and its growth is possible only under specific conditions. A thermodynamic study on boron incorporation into GaN showed that the alloy can spontaneously decompose locally [52].

In summary, we have demonstrated the complex nature of boron-containing nitride alloys in terms of their specific bandgap behavior and the metastable character of their crystallographic structures. Despite the observed existence of the hexagonal phase at very low and very high *x* values, as well as for certain clustered atomic arrangements across the composition range, stability calculations restrict the hexagonal phase to high *x* values, in agreement with available experimental data. These findings may help in identifying the conditions under which the preferred hexagonal structure of these alloys can be realized. However, to fully confirm our conclusions and suggestions, significant progress in the epitaxial growth of boron-containing nitride thin films is necessary.

## Figures and Tables

**Figure 1 materials-17-05120-f001:**
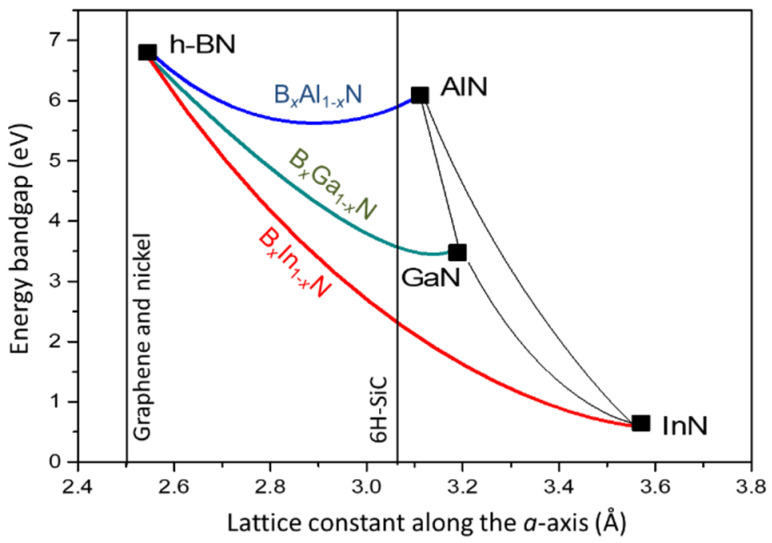
Schematic illustration of the bandgap dependence on lattice constant/alloy composition for B*_x_*Al_1−*x*_N, B*_x_*Ga_1−*x*_N, and B*_x_*In_1−*x*_N alloys. The values of h-BN, AlN, GaN, and InN bandgaps and lattice constants are taken from the experimental data given in Table 1; bowings are schematical.

**Figure 2 materials-17-05120-f002:**
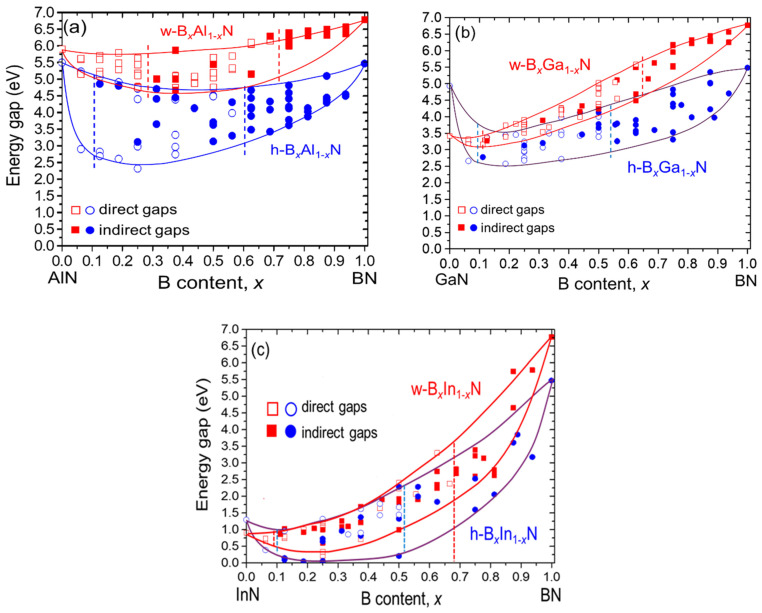
B*_x_*Al_1−*x*_N (**a**), B*_x_*Ga_1−*x*_N (**b**), and B*_x_*In_1−*x*_N (**c**) bandgap dependence on boron content in the wurtzite (w) and hexagonal (h) structures. Red squares and blue circles correspond to w and h structures, respectively. Open symbols correspond to direct bandgaps, and filled symbols to indirect bandgaps. See text for more details.

**Figure 3 materials-17-05120-f003:**
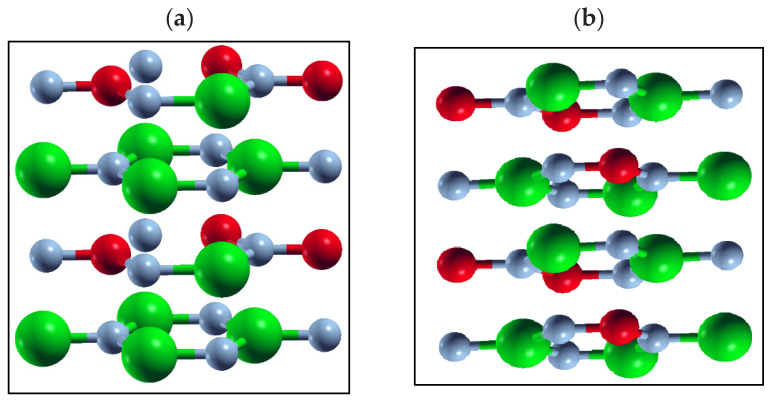
The atomic structure for the B*_x_*Al_1−*x*_N, *x* = 0.375, corresponds to (**a**) the lowest bandgap value (2.75 eV) and (**b**) the higher bandgap value (2.98 eV) for the same composition. Al atoms are large green balls. B atoms are smaller red balls and N is represented by the smallest gray balls.

**Figure 4 materials-17-05120-f004:**
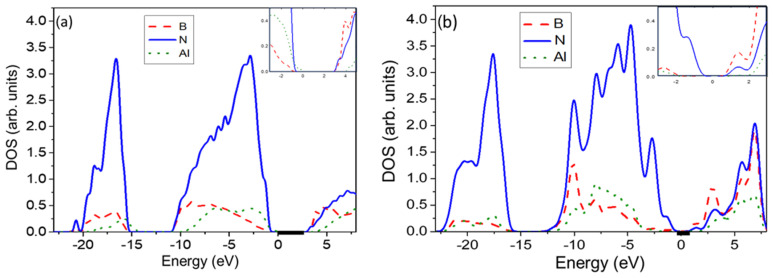
Partial DOS for the h-B*_x_*Al_1−*x*_N, (*x* = 0.375) in two cases: (**a**) uniform boron distribution (Eg = 4.46 eV) and (**b**) clustered B distribution (Eg = 2.75 eV). Zoom-in plots of the top of the valence band and the bottom of the conduction band are included. Bold lines indicate the bandgaps.

**Figure 5 materials-17-05120-f005:**
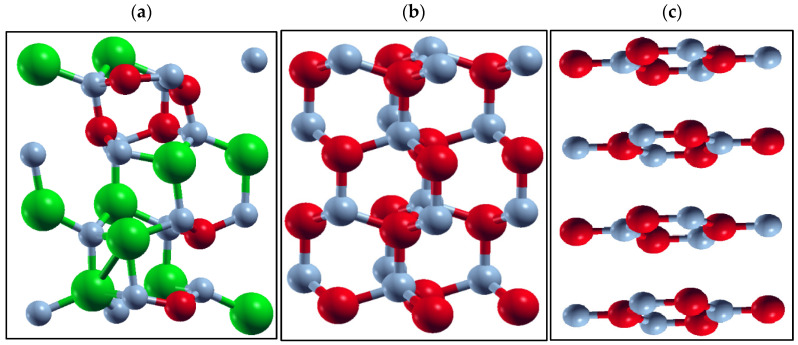
The crystallographic structure corresponding to (**a**) the highest point in an *x* = 0.375 set of the h-B*_x_*Al_1−*x*_N bandgap values (E_g_ = 4.46 eV) shown in Figure 3a, (**b**) wurtzite BN, and (**c**) hexagonal BN. Al atoms are large green balls. B atoms are smaller red balls and nitrogen is represented by the smallest gray balls.

**Figure 6 materials-17-05120-f006:**
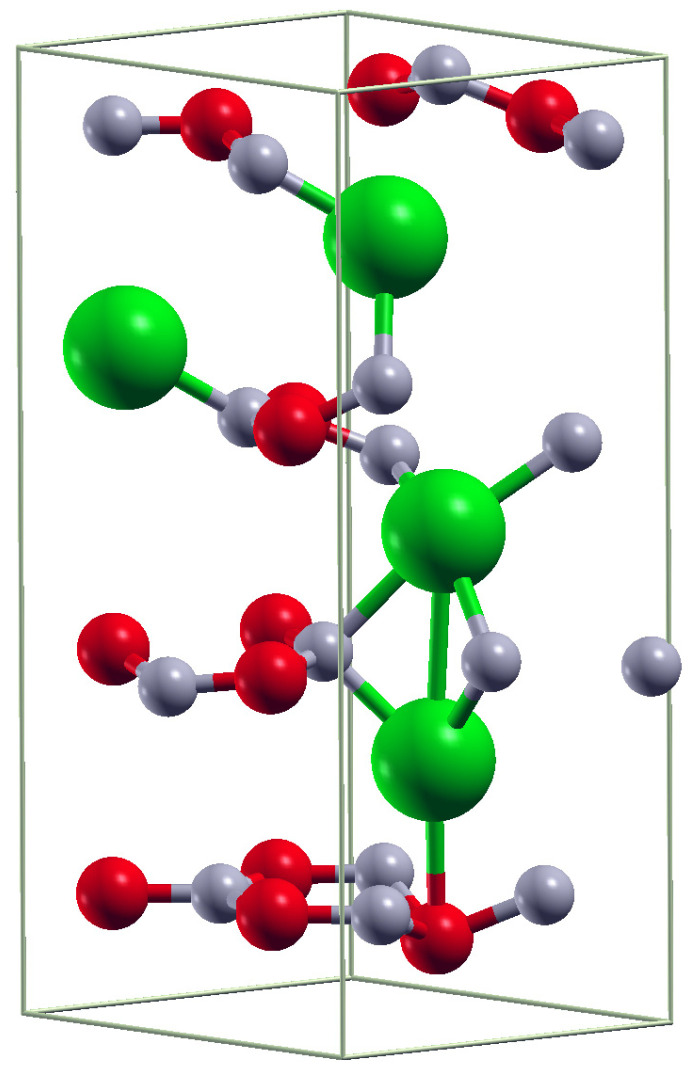
The crystallographic structure of the B*_x_*In_1−*x*_N alloy with *x* = 0.75 and E_g_ = 2.53 eV. Indium atoms are large green balls, boron atoms are smaller red balls, and nitrogen is represented by the smallest gray balls.

**Figure 7 materials-17-05120-f007:**
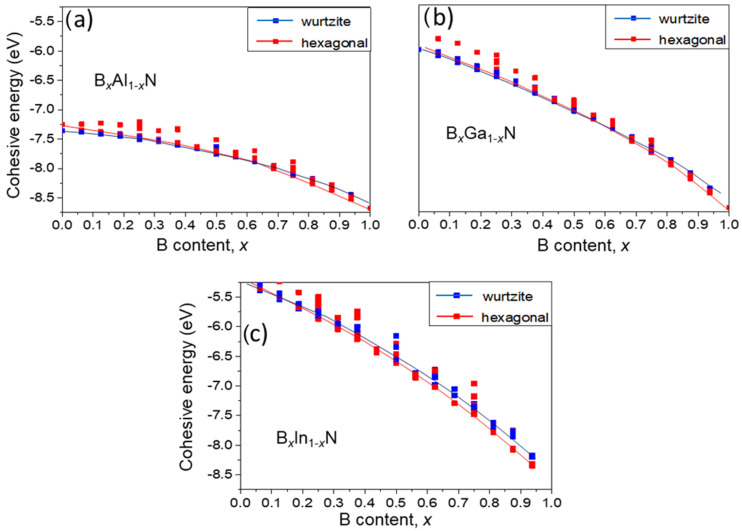
Calculated cohesive energies for three considered alloys. Solid red and blue lines fit to the lowest hexagonal and wurtzite points, respectively: (**a**) B*_x_*Al_1−*x*_N, (**b**) B*_x_*Ga_1−*x*_N, (**c**) B*_x_*In_1−*x*_N.

**Table 1 materials-17-05120-t001:** Lattice parameters and bandgap values for the nitrides investigated in the wurtzite structure in comparison to the available experimental data.

	Lattice Parameter a (Å)		Lattice Parameter c (Å)		Bandgaps E_g_ (eV)	
	DFT	Experiment	DFT	Experiment	DFT + HSE	Experiment
h-BN	2.50	2.504 [11]	6.36	6.6612 [11]	5.48	6.08 [3]
AlN	3.1126	3.111 [12]	4.9815	4.981 [12]	6.19	6.09 [13]
GaN	3.1955	3.1890 [14]	5.2040	5.1864 [14]	3.41	3.47 [15,16]
InN	3.5705	3.5705 [17]	5.7418	5.703 [17]	0.90	0.65 [18,19,20]

## Data Availability

The original contributions presented in the study are included in the article, further inquiries can be directed to the corresponding author.

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
