# Peer review of "Bandgap Characteristics of Boron-Containing Nitrides—Ab Initio Study for Optoelectronic Applications"

_materials, 2024, doi:10.3390/ma17205120_

Round 1
Reviewer 1 Report
Comments and Suggestions for Authors
The manuscript reports first principle calculations for boron containing nitride alloys. The paper notes the complex nature of the material family. The conclusion is that considerable effort is needed from experimental work in the area of thin-film fabrication to confirm the conclusions and suggestions.
The paper could be accepted after a revision. I recommend a thorough revision of the introduction and discussion sections for statements where relevant citations could be added.
Other comments/questions:
1. There are places where more citations are needed. For example, lines 66-74 makes many statements without any references. Please add appropriate citations to those statements. They are forming part of the problem this work is addressing.
2. It is not clear how the data for Fig. 1 was obtained. Are these from a citable source, or calculated/experimentally obtained? Please clarify.
3. Lines 91-96. Please cite sources for evidence.
4. Lines 121-122: Citation needed.
5. The figure 2 scales could be adjusted to add clarity.
Author Response
Report 1
The manuscript reports first principle calculations for boron containing nitride alloys. The paper notes the complex nature of the material family. The conclusion is that considerable effort is needed from experimental work in the area of thin-film fabrication to confirm the conclusions and suggestions.
The paper could be accepted after a revision. I recommend a thorough revision of the introduction and discussion sections for statements where relevant citations could be added.
Other comments/questions:
- There are places where more citations are needed. For example, lines 66-74 makes many statements without any references. Please add appropriate citations to those statements. They are forming part of the problem this work is addressing.
Response
We have added citations to the text (new lines 72-77):
“Despite the growing interest in h-BN and the reported experimental progress [9,10], there remains limited theoretical exploration of this material. Even the nature of the bandgap in h-BN alloys is still a subject of debate [3,4]. However, supporting ongoing experimental work in laboratories worldwide with a solid theoretical foundation is crucial for advancing the application of quantum structures based on h-BN in deep UV optoelectronics.”
- It is not clear how the data for Fig. 1 was obtained. Are these from a citable source, or calculated/experimentally obtained? Please clarify.
Response
We have added the following explanation to the caption of Figure 1:
“The values of h-BN, AlN, GaN and InN band gaps and lattice constants are taken from the experimental data given in Table 1, bowings are schematical.”
Lines 91-96. Please cite sources for evidence.
Response
We have added citations to the text (new lines 127-133):
“The existing theoretical and experimental data indicate a highly complex nature of h-BAlN and h-BGaN alloys concerning their bandgaps and crystallographic structures. Moreover, the reported data vary significantly, suggesting different composition ranges where the hexagonal phase is stable [18-22]. Therefore, we have decided to conduct a comprehensive study of the bandgap behavior of all three alloys— BxAl1-xN, BxGa1-xN, and BxIn1-xN —in both hexagonal and wurtzite phases, including various atomic arrangements within the unit cell.
- Lines 121-122: Citation needed.
Response
We have added a new reference 29 (new lines 163-165):
“The standard ab initio method results for semiconductors tend to be inaccurate, as they typically underestimate energy bandgaps by about 40% compared to experimental observations [29].”
[29] Poul Georg Moses, Maosheng Miao, Qimin Yan, Chris G Van de Walle “Hybrid functional investigations of band gaps and band alignments for AlN, GaN, InN, and InGaN”, J. Chem. Phys. 134 (2011) 084703.
- The figure 2 scales could be adjusted to add clarity.
Response
We made it.

Reviewer 2 Report
Comments and Suggestions for Authors
The authors reported ab-initio study on boron-containing nitrides on the band gaps focusing on hexagonal and wurtzite crystal structures. Their results show that for boron content larger than 0.85, the structures are hexagonal. The band gap is only direct when boron content smaller than 0.1, and becomes only indirect when boron content is greater than 0.7. The results will contribute to two-dimensional nitride research and future flexible optical device development. The investigations are sound, and the results are clear presented. As for the manuscript, minor revision is required before it can be published.
Comments
1. Line 27: Lack of unit for the temperature.
2. Line 169-171: The x1 and x2 are better using subscripts for the numbers.
3. For Figure 4: It would be better to have a zoom-in plot for the marked valence band edge to see the difference between both conditions. In addition, it would be also nice to have dashed lines to mark the band gap for better understanding.
4. Line 304 and 366-375: Please remove the repeated sentences.
Author Response
Report 2
The authors reported ab-initio study on boron-containing nitrides on the band gaps focusing on hexagonal and wurtzite crystal structures. Their results show that for boron content larger than 0.85, the structures are hexagonal. The band gap is only direct when boron content smaller than 0.1, and becomes only indirect when boron content is greater than 0.7. The results will contribute to two-dimensional nitride research and future flexible optical device development. The investigations are sound, and the results are clear presented. As for the manuscript, minor revision is required before it can be published.
Comments
- Line 27: Lack of unit for the temperature.
Response
We have added the temperature unit in the new line 30:
“niobium diselenide, a 2D superconductor with a critical temperature TC =7.2 K”
- Line 169-171: The x1 and x2 are better using subscripts for the numbers.
Response
We have done it (new lines 225-226):
“from 0 to x1, from x1 to x2, and from x2 to 1. In w-BxAl1-xN x1~0.29, x2~0.72, in w-BxGa1-xN x1~0.12, x2~0.65, and in w-BxIn1-xN x1~0.10, x2~0.68.”
- For Figure 4: It would be better to have a zoom-in plot for the marked valence band edge to see the difference between both conditions. In addition, it would be also nice to have dashed lines to mark the band gap for better understanding.
Response
Thank you very much. Thanks to your comment, we realized that Fig. 4a and Fig. 4b are identical. This was our mistake. Now we have corrected it and added zoom-in plots of the top of the valence band and the bottom of the conduction band, as suggested by you.
We have added the following explanation to the caption of Figure 4:
“Zoom-in plots of the top of the valence band and the bottom of the conduction band are included. Bold lines indicate the band gaps.”
- Line 304 and 366-375: Please remove the repeated sentences.
Response
Thank you. We have removed them (see new lines 367 and 436-443).

Reviewer 3 Report
Comments and Suggestions for Authors
This manuscript presents a study of the bandgap dependences of boron containing nitrides. It is modeling, ab initio study with focus on optoelectronic applications. Such a study of 2D group III nitrides with boron content is necessary and the authors adopt an appropriate level of theory and approach to it. The results are credible and corroborate works by others. It can help a broad spectrum of theoretical work in 2D group IIIA nitrides as well as other 2D semiconductors.
Some trends and references are missed and there are minor imprecisions in presenting aspects and details of methods and results. implying a minor revision before the acceptance of this much needed review for publication:
1: It is incorrect in English to begin each word in a title with a capital letter
2: The abstract should be more explicit and concrete in terms of level of theory adopted for modelling.
3: Most important compositions such as InGaN should be enumerated and mentioned early in the introduction. Many readers would like to have the material systems of main interest explicitly and early in the text flow.
4: The authors should be aware and should mention some seminal works based on ab initio simulations (incl. band gap evaluation) of ternary III-V compounds especially because of their close corroboration of supporting experimental developments, e.g., M.A.M. Filho et al., ACS Nanosci. Au 2023, 3, 1, 84–93; and Cryst Growth Des. 2024; 24(11): 4717–4727.
5: The authors just mention growth of h-BGaN by MOCVD but it is necessary to make clear the importance of the MOCVD technique to a vast majority of all 2D group III nitrides.
6: In some cases, ternary group IIIA nitrides can be unstable due to immiscibility issues, including in 2D. This aspect should be commented on.
7: The manuscript is well-written; however, it would still benefit from stylistic and grammatical revision.
Author Response
Report 3
This manuscript presents a study of the bandgap dependences of boron containing nitrides. It is modeling, ab initio study with focus on optoelectronic applications. Such a study of 2D group III nitrides with boron content is necessary and the authors adopt an appropriate level of theory and approach to it. The results are credible and corroborate works by others. It can help a broad spectrum of theoretical work in 2D group IIIA nitrides as well as other 2D semiconductors.
Some trends and references are missed and there are minor imprecisions in presenting aspects and details of methods and results. implying a minor revision before the acceptance of this much needed review for publication:
1: It is incorrect in English to begin each word in a title with a capital letter
Response
We have corrected it.
2: The abstract should be more explicit and concrete in terms of level of theory adopted for modelling.
Response
We have improved the abstract:
Abstract: Hexagonal boron nitride (h-BN) is recognized as a 2D wide bandgap material with unique properties, such as effective photoluminescence and diverse lattice parameters. Nitride alloys containing h-BN have the potential to revolutionize the electronics and optoelectronics industries. The energy band structures of three boron-containing nitride alloys - BxAl1-xN, BxGa1-xN, and BxIn1-xN - were calculated using standard density functional theory (DFT) with the hybrid Heyd-Scuseria-Ernzerhof (HSE) functional to correct lattice parameters and energy gaps. The results for both wurtzite and hexagonal structures reveal several notable characteristics, including a wide range of bandgap values, the presence of both direct and indirect bandgaps, and phase mixing between wurtzite and hexagonal structures. The hexagonal phase in these alloys is observed at very low and very high boron concentrations (x), as well as in specific atomic configurations across the entire composition range. However, cohesive energy calculations show that the hexagonal phase is more stable than the wurtzite phase only when x > 0.5, regardless of atomic arrangement. These findings provide practical guidance for optimizing the epitaxial growth of boron-containing nitride thin films, which could drive future advancements in electronics and optoelectronics applications.
3: Most important compositions such as InGaN should be enumerated and mentioned early in the introduction. Many readers would like to have the material systems of main interest explicitly and early in the text flow.
Response
We now explicitly mention the compositions of the alloys studied early in the introduction.
In lines 78-79:
“To address this need, we will focus on the electronic band structure of hexagonal alloys— h-BxAl1-xN, h-BxGa1-xN, and h-BxIn1-xN —across the full range of compositions (x).”
and at lines 130-133:
“Therefore, we have decided to conduct a comprehensive study of the bandgap behavior of all three alloys— BxAl1-xN, BxGa1-xN, and BxIn1-xN —in both hexagonal and wurtzite phases, including various atomic arrangements within the unit cell.”
We have also added a new part of text at the end of the introduction (lines 134-138):
“In the next section, we present the methods of calculation. Section 3 provides a description of the results and discussion, focusing on direct and indirect bandgaps, the wide range of their values, and significant bowings. We also examine the effects of different atomic arrangements (clustering) and the impact of hexagonal-wurtzite phase mixing. Section 4 summarizes our findings.”
4: The authors should be aware and should mention some seminal works based on ab initio simulations (incl. band gap evaluation) of ternary III-V compounds especially because of their close corroboration of supporting experimental developments, e.g., M.A.M. Filho et al., ACS Nanosci. Au 2023, 3, 1, 84–93; and Cryst Growth Des. 2024; 24(11): 4717–4727.
Response
We have added the following text at the beginning of the 'Stability of the Structures' section (lines 409-411):
“One of the most fundamental properties of crystalline systems is cohesive energy, which plays a crucial role in the formation of structures such as 2D layered materials and nanorods [49,50].”
In this text, we have incorporated two new references suggested by the referee:
[49] Manoel Alves Machado Filho, Ching-Lien Hsiao, Renato Batista dos Santos, Lars Hultman, Jens Birch, and Gueorgui K. Gueorguiev, “Self-Induced Core−Shell InAlN Nanorods: Formation and Stability Unraveled by Ab Initio Simulations” ACS Nanosci. Au (2023), 3, 84−9.
[50] Manoel Alves Machado Filho , William Farmer , Ching-Lien Hsiao , Renato Batista Dos Santos , Lars Hultman , Jens Birch , Kumar Ankit , Gueorgui Kostov Gueorguiev, “Density Functional Theory-Fed Phase Field Model for Semiconductor Nanostructures: The Case of Self-Induced Core-Shell InAlN Nanorods” Cryst Growth Des. (2024) 24, 4717-4727.
5: The authors just mention growth of h-BGaN by MOCVD but it is necessary to make clear the importance of the MOCVD technique to a vast majority of all 2D group III nitrides.
6: In some cases, ternary group IIIA nitrides can be unstable due to immiscibility issues, including in 2D. This aspect should be commented on.
Response to points 5 and 6
In response to the above criticisms/suggestions, we have added the following text (lines 102–122), along with new references [11–17]:
“The significant lattice mismatch between BN and AlN, as well as GaN, leads to large miscibility gaps. As a result, these alloys have predominantly been studied in the dilute regime of boron incorporation, owing to the inherent difficulties in achieving a uniform mixture across the full range of boron content.
The MOVPE (metal-organic vapor phase epitaxy) technique remains the most successful growth method for classical nitrides, so it is not surprising that it has been applied to the growth of BAlN and BGaN alloy systems. Despite the large lattice mismatch between BN and AlN, secondary-ion mass spectrometry (SIMS) measurements have confirmed the incorporation of up to 17% aluminum in layered BAlN alloy systems [11]. The same group was even able to produce Distributed Bragg Reflectors [12]. However, other reports using the same technique have shown significantly lower aluminum concentrations [13]. For the BGaN system, the highest reported gallium concentration, up to 7%, was documented by Wang in his PhD thesis [14].
High boron doping in GaN and AlN layers is a more common approach, and considerable efforts have been made in this area. The highest reported boron concentration for BAlN films was by Polyakov et al. [15], who found that single-phase wurtzite (WZ) BAlN films could only be grown with boron compositions not exceeding 10%. The same author also reported relatively high boron concentrations, up to 7%, in BGaN layers [16]. Progress related to increasing boron concentrations in GaN has been thoroughly described by Możdżyńska et al. [17], while both BAlN and BGaN alloy systems, along with their stability issues, are covered in a comprehensive review by Kudrawiec et al. [18].
[11] X. Li, S. Sundaram, Y. El Gmili, F. Genty, S. Bouchoule, G.Patriache, P. Disseix, F. Réveret, J. Leymarie, J.-P. Salvestrini, R. D. Dupuis, P. L. Voss, and A. Ougazzaden, "Single crystalline boron rich B(Al)N alloys grown by MOVPE” J. Cryst. Growth 414, 119 (2015).
[12] M. Abid, T. Moudakir, G. Orsal, S. Gautier, A. En Naciri, Z. Djebbour, J.-H. Ryou, G. Patriarche, L. Largeau, H. J. Kim, Z. Lochner, K. Pantzas, D. Alamarguy, F. Jomard, R. D. Dupuis, J.-P. Salvestrini, P. L. Voss, and A. Ougazzaden “Distributed Bragg reflectors based on diluted boron-based BAlN alloys for deep ultraviolet optoelectronic applications” Appl. Phys. Lett. 100, 051101 (2012)
[13] J. IwaÅ„ski, M. Tokarczyk, A. K. DÄ…browska, J. PawÅ‚owski, P. Tatarczak, J. Binder, A. WysmoÅ‚ek ”Bandgap manipulation of hBN by alloying with aluminum: absorption properties of hexagonal BAlN “https://arxiv.org/abs/2305.15810
[14] Q. Wang “Epitaxial growth and characterization of hexagonal boron gallium nitride semiconductor alloys” PhD Dissertation
[15] A. Y. Polyakov, M. Shin, W. Qian, M. Skowronski, D. W. Greve, and R. G.Wilson, “Growth of AlBN solid solutions by organometallic vapor-phase epitaxy,” J. Appl. Phys. 81, 1715 (1997).
[16] A.Y. Polyakov , M. Shin, M. Skowronski, D. W. Greve, R. G.Wilson, A.V. Govorkov and R.M. Desrosiers “Growth of GaBN ternary solutions by organometallic vapor phase epitaxy” J Electron Mater 26:237–242. (1997)
[17] E. B. MożdżyÅ„ska, S. ZÅ‚otnik, P. Ciepielewski, J. Gaca, M. Wójcik , P. P. MichaÅ‚owski, K. RosiÅ„ski, K. PiÄ™tak, M. RudziÅ„ski, E. Jezierska, and J. M. Baranowski „Insights on boron impact on structural characteristics in epitaxially grown BGaN “ J Mater Sci (2022) 57:7265–7275
7: The manuscript is well-written; however, it would still benefit from stylistic and grammatical revision.
Response
We have improved the style and grammar throughout the entire manuscript.

Round 2
Reviewer 1 Report
Comments and Suggestions for Authors
The manuscript has been revised and the questions/comments from the review have been addressed. The paper could be accepted for publication.
Reviewer 2 Report
Comments and Suggestions for Authors
The authors reported ab-initio study on boron-containing nitrides on the band gaps focusing on hexagonal and wurtzite crystal structures. Their results show that for boron content larger than 0.85, the structures are hexagonal. The band gap is only direct when boron content smaller than 0.1, and becomes only indirect when boron content is greater than 0.7. The results will contribute to two-dimensional nitride research and future flexible optical device development. The investigations are sound, and the results are clear presented.
In this revised manuscript, the abstract is clearer, more information about the three major nitrides are provided for the readers to have an better overall picture of this study. The writing mistakes are removed or corrected with careful text editing. Therefore, I recommend this manuscript for publication.
Reviewer 3 Report
Comments and Suggestions for Authors
The authors have comprehensively revised the manuscript. I fully support the publication of this revised manuscript